# A Neural Network Monte Carlo Approximation for Expected Utility Theory

**Yichen Zhu † and Marcos Escobar-Anel *,†** 🔟

Department of Statistical and Actuarial Sciences, Western University, London, ON N6A5B7, Canada; yzhu562@uwo.ca
* Correspondence: marcos.escobar@uwo.ca
† These authors contributed equally to this work.

**Abstract:** This paper proposes an approximation method to create an optimal continuous-time portfolio strategy based on a combination of neural networks and Monte Carlo, named NNMC. This work is motivated by the increasing complexity of continuous-time models and stylized facts reported in the literature. We work within expected utility theory for portfolio selection with constant relative risk aversion utility. The method extends a recursive polynomial exponential approximation framework by adopting neural networks to fit the portfolio value function. We developed two network architectures and explored several activation functions. The methodology was applied on four settings: a 4/2 stochastic volatility (SV) model with two types of market price of risk, a 4/2 model with jumps, and an Ornstein–Uhlenbeck 4/2 model. In only one case, the closed-form solution was available, which helps for comparisons. We report the accuracy of the various settings in terms of optimal strategy, portfolio performance and computational efficiency, highlighting the potential of NNMC to tackle complex dynamic models.

**Keywords:** neural networks; expected utility theory; CRRA utility; 4/2 stochastic volatility model

## 1. Introduction

Optimally allocating a collection of financial investments such as stocks, bonds and commodities has been a topic of concern to financial institutions and shareholders at least since the pioneering work of Markowitz's mean-variance portfolio theory in 1952. People then realized the potential of diversification and their work laid the foundations for the development of portfolio analysis in both academia and industry. These initial results were in discrete-time, but it was not long before continuous-time portfolio decisions were produced in the alternative paradigm of expected utility theory, as can be seen in Merton (1969). The author assumed that the investor is able to continuously adjust their position, and the stock price process is modelled by a geometric Brownian motion (GBM). The optimal trading strategy and consumption policy that maximize the investor's expected utility were obtained in closed-form by solving a Hamilton–Jacobi–Bellman equation.

The beauty and practicality of this continuous-time solution has led many researchers onto this path, producing optimal closed-form strategies for a wide range of models. For example, Kraft (2005) considered the stochastic volatility (SV) Heston model, Heston (1993). Flor and Larsen (2014) constructed a portfolio of stocks and fixed-income market products to hedge the interest rate risk. Explicit solutions in the presence of regime switching, stochastic interest rate and stochastic volatility was presented in Escobar et al. (2017), whilst the positive performance of their portfolio is confirmed by empirical study. For the commodities asset class, Chiu and Wong (2013) modelled a mean-reverting risky asset by an exponential Ornstein–Uhlenbeck (OU) process and solved the investment problem for an insurer subject to the random payment of insurance claim.

These models are particular cases of the quadratic-affine family (see Liu (2006)), one of the broadest models solvable in closed-form. The value function for a model in

this family is the product of a function of wealth and an exponential quadratic function. Nonetheless, the complexity of financial markets has continued increasing every decade, with researchers detecting new stylized facts and proposing new models outside the quadratic-affine. Needless to say, investors must rely on these advanced models for better financial decisions, however. closed-form solutions are no longer guaranteed. One example of these advanced models is the GBM 4/2 model, introduced in Grasselli (2017). The model improves the Heston model in terms of the better fitting of implied volatility surfaces and historical volatilities patterns. The optimal portfolio problem with the GBM 4/2 model is solvable for certain types of market price of risk (MPR, see Cheng and Escobar-Anel (2021)), while the optimal trading strategy has not been found yet with an MPR proportional to the instantaneous volatility. More recently, an OU 4/2 model, which unifies the mean-reverting drift and stochastic volatility in a single model, was presented in Escobar-Anel and Gong (2020). The model targets two asset classes: commodities and volatility indexes. The optimal portfolio with the OU 4/2 model is not in closed form. This motivates approximation methods for dynamic portfolio choice.

Most approximation methods follow the idea from martingale method (see Karatzas et al. (1987)) or dynamic programming technique Brandt et al. (2005). Cvitanić et al. (2003) proposed a simulation-based method seeking the financial replication of the optimal terminal wealth given in the martingale method. Detemple et al. (2003) developed a comprehensive approach for the same investment problems, and the application of Malliavin calculus enhances its accuracy. The work in Brandt et al. (2005) led to the BGSS method, which was inspired by the popular least-square Monte Carlo method of Longstaff and Schwartz (2001). BGSS pioneered the recursive approximation method for dynamic portfolio choice. Cong and Oosterlee (2017) enhanced BGSS with the stochastic grid bundling method (SGBM) for conditional expectation estimation introduced in Jain and Oosterlee (2015). More recently, a polynomial affine method for constant relative risk aversion utility (PAMC) was recently developed in Zhu et al. (2020). The method takes advantage of the quadratic-affine structure, leading to superior accuracy and efficiency in the approximation of the optimal strategy and value function. In this paper, we extend the methodology in PAMC using neural networks.

The history of artificial neural networks goes back to McCulloch and Pitts (1943), where the author created the so-called "threshold logic" on the basis of the neural networks of the human brain in order to mimic human thoughts. Deep learning has since steadily evolved. Almost three decades later, back propagation, a widely used algorithm in neural network's parameter fitting for supervised learning, was introduced, see Linnainmaa (1970). The importance of back propagation was only fully recognized when Rumelhart et al. (1986) showed that it can provide interesting distribution representations. The universal approximation theorem (see Cybenko (1989)) illustrated that every bounded continuous function can be approximated by a network with an arbitrarily small error, which further verifies the effectiveness of the neural network. Neural networks recently attracted a lot of attention of applied scientists, and were successful in fields such as image recognition and natural language processing because they are particular good at function approximation when the form of the target function is unknown. In the realm of dynamic portfolio analyses, Lin et al. (2006) first predicted portfolio covariance matrix with the Elman network and achieved the good estimation of the optimal mean-variance portfolio. More recently, Li and Forsyth (2019) proposed a neural network, representing the portfolio strategy at each rebalancing time, for a constrained defined contribution (DC) allocation problem. Chen and Ge (2021) introduced a differential equation-based method, where the value function with the Heston model is estimated by a deep neural network.

In this paper, motivated by the lack of knowledge on the correct expression for the portfolio value function for unsolvable models, we approximated the optimal portfolio strategy for any given stochastic process model with a neural network fitting the value function. Successful fitting relies on a suitable network architecture that captures the connection between input and output variables, as well as reasonable activation functions.

We designed two architectures enriching an embedded quadratic-affine structure, and we considered three types of activation functions.

Given the lack of closed-form solutions for SV 4/2 models, we used them as our toy examples in the implementations. In particular, we first implemented our methodology in the solvable case (i.e., GBM 4/2 with solvable MPR), so the accuracy and efficiency were demonstrated before it is applied to the unsolvable cases of: GBM 4/2 model with stochastic jumps, GBM 4/2 model with proportional instantaneous volatility MPR, and the OU 4/2 model. Furthermore, we numerically show which network architecture is preferable in each case.

The paper is organized as follows. Section 2 introduces the dynamic portfolio choice problem, and presents the neural network architectures, activation functions and parameter training details. The step-by-step algorithm of our methodology is provided in Section 3. Sections 4 and 5 apply the methodology to the GBM 4/2 and the OU 4/2 models. Section 6 concludes.

## 2. Problem Setting and Architectures of the Deep Learning Model

We considered a frictionless market consisting of a money market account (cash, *M*) and one stock (*S*). We assume the stock price follows a generalized diffusion process incorporating a one-dimensional state variable *X*. All the processes are defined on a complete probability space $(\Omega, \mathcal{F}, \mathbb{P})$ with a right-continuous filtration $\{\mathcal{F}_t\}_{t \in [0,T]}$, summarized by the stochastic differential equations (SDE):

$$
\begin{cases}
\frac{dM_t}{M_t} = r(X_t)dt \\
dS_t = S_t\theta(X_t, S_t)dt + S_t\sigma(X_t, S_t)dB_t + S_{t-}\mu_N dN_t \\
dX_t = a(X_t)dt + b(X_t)dB_t^X \\
< dB_t, dB_t^X > = \rho dt.
\end{cases}
\tag{1}
$$

$B_t$ and $B_t^X$ are Brownian motions with correlation $\rho$. $r(X_t)$ is the interest rate, $\theta(X_t, S_t)$ and $\sigma(X_t, S_t)$ are the drift and diffusion coefficients for the stock price. $a(X_t)$ and $b(X_t)$ are measurable functions of state variable $X_t$. $N_t$ is a pure-jump process independent of $B_t$ and $B_t^X$ with stochastic intensity $\lambda_N X_t$ for constant $\lambda_N > 0$, and $\mu_N > -1$ denotes the jump size.

We consider an investor with risk preference represented by a constant relative risk aversion (CRRA) utility:

$$
U(W) = \frac{W^{1-\gamma}}{1-\gamma}.
\tag{2}
$$

Investors can adjust their allocation at a predetermined set of rebalancing times $(0, \Delta t, 2\Delta t, ..., T - \Delta t)$. The investors wish to derive a portfolio strategy $\pi$ (percentage of wealth allocated to the stock) that maximizes their expected utility of terminal wealth, in other words, $\mathbb{E}(U(W_T))$. The value function, representing the investor's conditional expected utility, has the following representation:

$$
V(t, W, S, X) = \max_{\pi_{s \geq t}} \mathbb{E}(U(W_T) \mid t, W, S, X) = \frac{W^{1-\gamma}}{1-\gamma} f(t, S, X).
\tag{3}
$$

The value function is separated into a wealth factor $\frac{W^{1-\gamma}}{1-\gamma}$ and a state variable function *f*. The NNMC estimates the state variable function *f* with a neural network model *NN* and computes the optimal strategy $\pi_t^*$ with the Bellman principle.

### 2.1. Architectures of the Deep Learning Model

In this section, we present two neural network architectures to fit the value function. According to the separable property of the value function shown in (3), the only unknown component is the state variable function *f*, which is therefore the target function for the

neural network. The architectures of the networks are built around exponential polynomial functions, which are the most common form of solvable investor's value functions and used in the PAMC method (see Zhu et al. (2020)). This property of proposed networks ensures that the new method generalizes PAMC.

The neural network is expected to achieve a better fit than a polynomial regression if the true state variable function is significantly different from the exponential polynomial function. Furthermore, we designed an initialization method for networks, which is better than a random initialization in terms of portfolio value function fitting.

### 2.1.1. Sum of Exponential Network

We first introduced the sum of the exponential polynomial neural network (SEN), as illustrated in Figure 1. The amount of input depends on the number of state variables. For simplicity, we took two inputs as an example. The first hidden layer computes the monomial of inputs. The second hidden layer obtains the linear combinations of the neuron in the first layer, where the weights are fitted in NNMC. An exponential activation function is applied to the second layer. The final output calculates a linear combination of exponential polynomials, so the exponential polynomial is a specific case of this neural network.

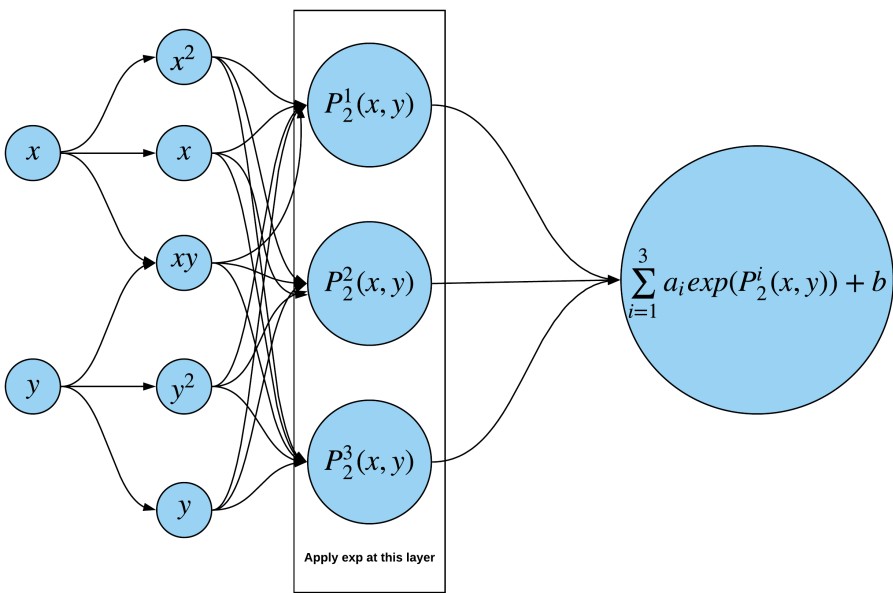

**Figure 1.** Sum of exponential network (SEN).

We denote the sum of exponential network by $NN^{SEN}$; the proposition next states the estimation of the corresponding optimal allocation.

**Proposition 1.** *Given the SEN approximation of the value function at the next rebalancing time $t + \Delta t$, (i.e., $NN^{SEN}[t + \Delta t, S_t, X_t]$), the optimal strategy at time $t$ is given by*

$$\pi_t^{SEN} = \arg\max_{\pi} V(t, W_t, \pi_t, S_t, X_t) \tag{4}$$

*which is the solution of:*

$$f_2(t, W_t, S_t, X_t) + f_1(t, W_t, S_t, X_t)\pi_t + NN^{SEN}(t + \Delta t, S_t(1 + \mu_N), X_t)\lambda_N X_t \mu_N (1 + \pi_t \mu_N)^{-\gamma} = 0, \tag{5}$$

*where:*

$$f_1(t, W_t, S_t, X_t) = -\gamma NN^{SEN}(t + \Delta t, W_t, S_t, X_t)\sigma^2(X_t, S_t)$$

$$
\begin{aligned}
f_2(t, W_t, S_t, X_t) = {} & NN^{SEN}(t + \Delta t, W_t, S_t, X_t)(\theta(X_t, S_t) - r(X_t))) \\
& + \frac{\partial NN^{SEN}(t + \Delta t, W_t, S_t, X_t)}{\partial S_t} S_t \sigma^2(X_t, S_t) \\
& + \frac{\partial NN^{SEN}(t + \Delta t, W_t, S_t, X_t)}{\partial X_t} \sigma(X_t, S_t)b(X_t)\rho.
\end{aligned}
$$

(6)

*Notably,* $\pi_t^{SEN} = -\frac{f_2(t, W_t, S_t, X_t)}{f_1(t, W_t, S_t, X_t)}$ *when* $S_t$ *follows a diffusion process, i.e.,* $\lambda_N = 0.$ $\pi_t^{SEN} = \frac{1}{\mu_N}((-\frac{f_2(t, W_t, S_t, X_t)}{NN^{SEN}(t + \Delta t, S_t, X_t)\lambda_N X_t \mu_N})^{-\frac{1}{\gamma}} - 1)$ *when* $S_t$ *follows a jump process, i.e.,* $\sigma(X_t, S_t) = 0.$

**Proof.** It follows similarly to Theorem 1 in Zhu and Escobar-Anel (2020). According to the Bellman principle:

$$V(t, W_t, S_t, X_t) = \max_{\pi_t} \mathbb{E}_t(V(t + \Delta t, W_{t+\Delta t}, S_{t+\Delta t}, X_{t+\Delta t}) \mid W_t, S_t, X_t).$$

(7)

We substitute $V(t + \Delta t, W_{t+\Delta t}, S_{t+\Delta t}, X_{t+\Delta t})$ with $\frac{W^{1-\gamma}}{1-\gamma}NN^{SEN}(t + \Delta t, W_{t+\Delta t}, S_{t+\Delta t}, X_{t+\Delta t})$ and expand the right hand side of the equation with respect to $W$, $S$ and $X$, then $V(t, W_t, S_t, X_t)$ is written as a function of strategy $\pi_t$. Equation (5) is obtained with the first order condition. $\square$

### 2.1.2. Improving Exponential Network

The architecture of an improving exponential network (IEN) is exhibited in Figure 2.

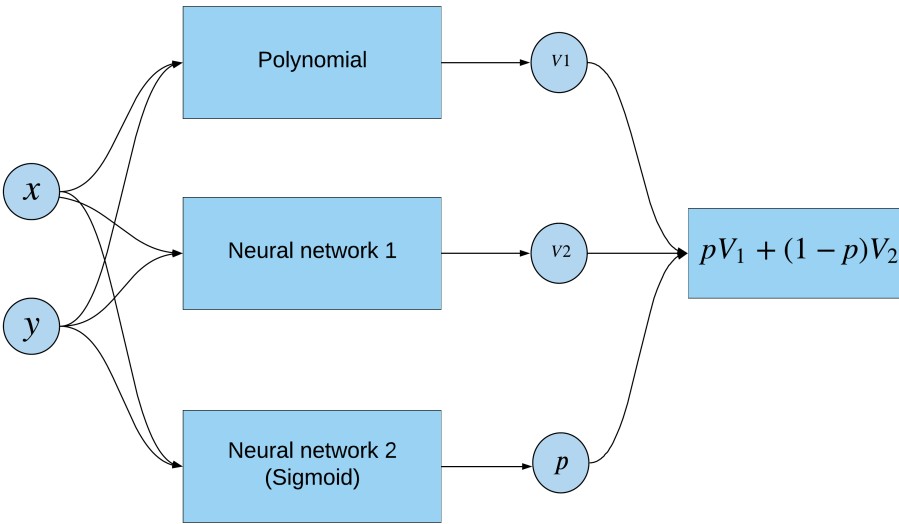

**Figure 2.** Improving exponential polynomial.

The target function of IEN is the log of the state variable function $f$ (i.e., $\ln f$). The neural network consists of three parts. Node 1 is a polynomial with the output denoted by $V_1$. Node 2 is an artificial neural network with an arbitrary number of hidden layers and neurons; we denoted its output by $V_2$. Node 3 is a single-layer network with a Sigmoid function which computes a proportion $p \in [0, 1]$. The final output is the weighted average of the first two nodes $pV_1 + (1-p)V_2$. The second node is the complement to the exponential polynomial function. Moreover, the similarity between the true value function and the exponential polynomial function is measured by $p$, which is fitted into the NNMC methodology. Therefore, the network automatically adjusts the weights on the exponential

polynomial function and its supplement according to the generated data. Finally, the state variable function $f$ is computed as

$$f = e^{pv_1 + (1-p)v_2} = (e^{v_1})^p \times (e^{v_2})^{1-p}, \tag{8}$$

which is the geometric weighted average of nodes 1 and 2. Letting $NN^{IEN}$ denote the IEN, the estimation of the optimal strategy is given in the next proposition.

**Proposition 2.** *Given the IEN approximation of the log value function at time $t + \Delta t$ (i.e., $NN^{IEN}[t + \Delta t, S_t, X_t]$), the optimal strategy at time $t$ is given by*

$$\pi_t^{IEN} = \arg\max_{\pi} V(t, W_t, S_t, X_t) \tag{9}$$

*which is the solution of:*

$$\left( f_2(t, W_t, S_t, X_t) + f_1(t, W_t, S_t, X_t)\pi_t \right) + \lambda_N X_t \exp\left( NN^{IEN}(t + \Delta t, S_t(1 + \mu_N), X_t) \right) \mu_N (1 + \pi_t \mu_N)^{-\gamma} = 0, \tag{10}$$

*where:*

$$
\begin{aligned}
f_1(t, W_t, S_t, X_t) &= -\gamma \exp\left( NN^{IEN}(t + \Delta t, S_t, X_t) \right) \sigma^2(X_t, S_t) \\
f_2(t, W_t, S_t, X_t) &= \exp\left( NN^{IEN}(t + \Delta t, S_t, X_t) \right) (\theta(X_t, S_t) - r(X_t))) \\
&\quad + \frac{\partial NN^{IEN}(t + \Delta t, W_t, S_t, X_t)}{\partial S_t} \exp\left( NN^{IEN}(t + \Delta t, S_t, X_t) \right) S_t \sigma^2(X_t, S_t) \\
&\quad + \frac{\partial NN^{IEN}(t + \Delta t, W_t, S_t, X_t)}{\partial X_t} \exp\left( NN^{IEN}(t + \Delta t, S_t, X_t) \right) \sigma(X_t, S_t) b(X_t) \rho.
\end{aligned}
\tag{11}
$$

*Notably, $\pi_t^{IEN} = -\frac{f_2(t, W_t, S_t, X_t)}{f_1(t, W_t, S_t, X_t)}$ when $S_t$ follows a diffusion process, in other words, $\lambda_N = 0$. $\pi_t^{IEN} = \frac{1}{\mu_N}\left(\left(-\frac{f_2(t, W_t, S_t, X_t)}{\exp\left(NN^{IEN}(t + \Delta t, S_t, X_t)\right) \lambda_N X_t \mu_N}\right)^{-\frac{1}{\gamma}} - 1\right)$ when $S_t$ follows a jump process (i.e., $\sigma[X_t, S_t] = 0$).*

**Proof.** The proof follows similarly to Proposition 1. □

*2.2. Initialization, Stopping Criterion and Activation Function*

In this section, we disclose more details on training the neural networks. The initialization of weights is the first step of network training, which may significantly impact the goodness of fit. A good initialization prevents the network's weights from converging to a local minimum and avoids slow convergence. Random initialization is mostly used as the interpretability of the network is usually weak. In contrast, both the SEN and the IEN are extensions of an exponential polynomial function; we suggest taking advantage of the results from the polynomial regression. Hence, the neural network searches the minimum near the exponential polynomial function used in the PAMC ensuring consistency. The polynomial regression initialization achieves superior results to the random initialization.

The coefficients of the exponential polynomial were first obtained with a regression model. The output of the SEN is a linear combination of exponential polynomial functions $\sum_{i=1}^{N} a_i exp(P_n^i(x, y)) + b$, we substitute the coefficients from polynomial regression into $P_n^1(x, y)$ and set $a_1 = 1, a_2 = a_3 = ... = a_n = b = 0$. For the initialization of the IEN, we substitute the coefficients into the first node and artificially make $p = 0$.

The training process minimizes the mean squared error (MSE) between the network's output and the simulated expected utility, and the sample data are split into a training set and a test set to reduce the overfitting problem. Adam is a back-propagation algorithm that combines the best properties of the AdaGrad and RMSProp algorithms to handle sparse

gradients on noisy problems and provides excellent convergence speed. We applied the Adam on the training set for updating the network's weights, and the test set MSE was computed and subsequently recorded. The test set MSE was expected to be convergent, so the training process was finished when the difference between the moving average of the recent 100 test set MSEs and the most recent test set MSE was less than a predetermined threshold, which was set at 0.00001 in the implementation.

The number of exponential polynomials is a hyperparameter in the SEN. We let the SEN be a sum of two exponential polynomial functions for simplicity. Node 2 in the IEN is an artificial neural network, which complements node 1 when the value function significantly deviates from an exponential polynomial function. The number of hidden layers and neurons, as well as the activation function of node 2, are freely determined before fitting the value function. We assume node 2 is a single layer network with 10 neurons and we implement several functions for comparison purposes, such as the logistic (sigmoid):

$$f(x) = \frac{1}{1 + e^{-x}}, \tag{12}$$

the Rectified linear unit (ReLU):

$$f(x) = \begin{cases} 0 & \text{if } x \leq 0 \\ x & \text{if } x > 0 \end{cases}, \tag{13}$$

and the Exponential linear unit (ELU):

$$f(x) = \begin{cases} 0 & \text{if } x \leq 0 \\ e^x - 1 & \text{if } x > 0 \end{cases} \tag{14}$$

## 3. Notation and Algorithm of the Methodology

In this section, we clarify the notation and the step-by-step algorithm. Table 1 displays a summary of the notation.

**Table 1.** Notation for NNMC is listed here.

| Notation | Meaning |
| --- | --- |
| $B_t^m$ | Brownian motion at time $t$ in $m_{th}$ simulated path |
| $S_t^m$ | Stock price at time $t$ in $m_{th}$ simulated path |
| $X_t^m$ | Other state variable such as interest rate or volatility |
| $n_r$ | Number of simulated paths |
| $N$ | Number of simulation to compute expected utility for a given set $(W_0, S_t^m, X_t^m)$ |
| $\hat{W}_{t+\Delta t}^{m,n}(\pi^m)$ | A simulated wealth level at $t + \Delta t$ given the wealth, allocation and other state variables at $t$ are $W_0$, $\pi^m$ and $X_t^m$ |
| $\hat{S}_{t+\Delta t}^{m,n}$ | A simulated stock price at $t + \Delta t$ given $S_t^m$ |
| $\hat{X}_{t+\Delta t}^{m,n}$ | A simulated state variable at $t + \Delta t$ give $X_t^m$ |
| $V(t, W, S, X)$ | Value function at time $t$ given wealth $W$, stock price $S$ and state variable $X$ |
| $NN(t, X, S)$ | The neural network used to fit $f(t, S_t, X_t)$ or $\ln[f(t, S_t, X_t)]$ |
| $\hat{v}^m$ | Estimation of $f(t, S_t^m, X_t^m)$ or $\ln[f(t, S_t^m, X_t^m)]$ |
| $\pi_s^{m,n}$ | Optimal strategy at time $s$ given wealth, stock price and other state variables are $\hat{W}_s^{m,n}$, $\hat{S}_s^{m,n}$ and $\hat{X}_s^{m,n}$ |
| $\hat{V}(0, W_0, S_0, X_0)$ | Estimation of expected utility at time 0. |

*Algorithm*

We first generated the paths of the stock price $S_t^m$ and state variable $X_t^m$. The method starts from $t = T - \Delta t$ (i.e., the last rebalancing time before the terminal). We computed the optimal strategy $\pi_{T-\Delta t}^m$ given $W_0$, $S_{T-\Delta t}^m$, $X_{T-\Delta t}^m$ using the Equation (5) or Equation (10). Then, $\hat{v}^m$ is obtained through simulation, which estimates $f(T - \Delta t, S_{T-\Delta t}^m, X_{T-\Delta t}^m)$ when using SEN and $\ln\left[f(T - \Delta t, S_{T-\Delta t}^m, X_{T-\Delta t}^m)\right]$ when using IEN. The network $NN(T - \Delta t, X, S)$, approximating the state variable function, is trained with the input $(X_{T-\Delta t}^m, S_{T-\Delta t}^m)$ and output $\hat{v}^m$. We conduct a similar procedure at each rebalancing point and recursively approximate the value function and optimal strategy until the inception of the portfolio. To evaluate the expected utility, we regenerated the paths of stock price and state variables. The path-wise optimal strategy was computed from $NN(t, X, S)$, so the optimal terminal wealth is easy to obtain. The average of the utility of optimal terminal wealth approximates the expected utility. Algorithms 1 and 2 present the pseudo code for NNMC using SEN and IEN, respectively. Simulation variance reduction methods, such as antithetic variates, could be incorporated into both algorithms to reduce the standard error of estimated expected utility.

---

**Algorithm 1:** NNMC-SEN

---

　　**Input:** $S_0, W_0, X_0$
　　**Output:** Optimal trading strategy $\pi_0^*$ and expected utility $\hat{V}(0, W_0, S_0, X_0)$

1　initialization;
2　Generating $n_r$ paths of $B_t^m, S_t^m, X_t^m$　　*for*　$m = 1 ... n_r$;
3　**while**　$t = T - \Delta t$ **do**
4　┃　Compute optimal allocation $\pi_{T-\Delta t}^m$ with Equation (5) ;
5　┃　Simulate wealth $\hat{W}_T^{m,n}(\pi_{T-\Delta t}^m)$ given $W_0$, $S_{T-\Delta t}^m$, $\pi_{T-\Delta t}^m$ and $X_{T-\Delta t}^m$ at $T - \Delta t$
　　┃　　*for*　$n = 1 ... N$;
6　┃　Compute $\hat{v}^m = \frac{1}{N} \sum\limits_{n=1}^{N} U(\hat{W}_T^{m,n}(\pi_{T-\Delta t}^m)) \times \frac{1-\gamma}{W_0^{1-\gamma}}$　　*for*　$m = 1 ... n_r$ ;
7　┃　Train a network with input $(X_{T-\Delta t}^m, S_{T-\Delta t}^m)$ and output $\hat{v}^m$. Denote the network
　　┗　　by $NN(T - \Delta t, X, S)$
8　**for** $t = T - 2\Delta t$ *to* $\Delta t$ **do**
9　┃　Compute optimal allocation $\pi_t^m$ with $NN(t + \Delta t, X, S)$ and Equation (5) given
　　┃　　$W_0, S_t^m$, and $X_t^m$ ;
10　┃　Simulate wealth $\hat{W}_{t+\Delta t}^{m,n}(\pi_t^m)$, $\hat{S}_{t+\Delta t}^{m,n}$ and $\hat{X}_{t+\Delta t}^{m,n}$ given $W_0, S_t^m, \pi_t^m$ and $X_t^m$ at
　　┃　　time $t$　　*for*　$n = 1 ... N$;
11　┃　Compute $\hat{v}^m = [\frac{1}{N} \sum\limits_{n=1}^{N} (W_{t+\Delta t}^{m,n}(\pi_t^m))^{1-\gamma} NN(t + \Delta t, \hat{X}_{t+\Delta t}^{m,n}, \hat{S}_{t+\Delta t}^{m,n})] \times \frac{1}{W_0^{1-\gamma}}$ *for*
　　┃　　$m = 1 ... n_r$ ;
12　┃　Train a new network with input $(X_{T-\Delta t}^m, S_{T-\Delta t}^m)$ and output $\hat{v}^m$ and denote it by
　　┗　　$NN(t, X, S)$ ;
13　**while**　$t = 0$ **do**
14　┃　Compute $\pi_0^*$ with with $NN(\Delta t, X, S)$ and Equation (5);
15　┃　Generate new paths of $S_t^z, X_t^z$　　*for*　$z = 1 ... N_0$, use the estimation of value
　　┃　　function $NN(t, X, S)$ to compute $\pi_t^z$ and $W_T^z$.
16　┃　The expected utility is, $\hat{V}(0, W_0, S_0, X_0) = \frac{1}{N_0} \sum\limits_{n=1}^{N_0} U(W_T^z)$
　　┗
17　**return** $\pi_0^*, \hat{V}(0, W_0, S_0, X_0)$

---

---

**Algorithm 2:** NNMC-IEN

---

**Input:** $S_0, W_0, X_0$
**Output:** Optimal trading strategy $\pi_0^*$ and expected utility $\hat{V}(0, W_0, S_0, X_0)$

**1** initialization;

**2** Generating $n_r$ paths of $B_t^m, S_t^m, X_t^m \quad for \quad m = 1...n_r$;

**3 while** $t = T - \Delta t$ **do**

**4** $\quad$ Compute optimal allocation $\pi_{T-\Delta t}^m$ with Equation (10);

**5** $\quad$ Simulate wealth $\hat{W}_T^{m,n}(\pi_{T-\Delta t}^m)$ given $W_0, S_{T-\Delta t}^m, \pi_{T-\Delta t}^m$ and $X_{T-\Delta t}^m$ at $T - \Delta t$
$\quad\quad$ *for* $\quad n = 1...N$;

**6** $\quad$ Compute $\hat{v}^m = \ln[sign(1 - \gamma)\frac{1}{N}\sum_{n=1}^{N} U(\hat{W}_T^{m,n}(\pi_{T-\Delta t}^m))] - (1 - \gamma)\ln[W_0]$
$\quad\quad$ *for* $\quad m = 1...n_r$;

**7** $\quad$ Train the network with input $(X_{T-\Delta t}^m, S_{T-\Delta t}^m)$ and output $\hat{v}^m$. Denote the
$\quad\quad$ network by $NN(T - \Delta t, X, S)$

**8 for** $t = T - 2\Delta t$ *to* $\Delta t$ **do**

**9** $\quad$ Compute optimal allocation $\pi_t^m$ with $NN(t + \Delta t, X, S)$ and Equation (10)
$\quad\quad$ given $W_0, S_t^m$, and $X_t^m$ ;

**10** $\quad$ Simulate wealth $\hat{W}_{t+\Delta t}^{m,n}(\pi_t^m)$, $\hat{S}_{t+\Delta t}^{m,n}$ and $\hat{X}_{t+\Delta t}^{m,n}$ given $W_0, S_t^m, \pi_t^m$ and $X_t^m$ at $t$
$\quad\quad$ *for* $\quad n = 1...N$;

**11** $\quad$ Compute

**12** $\quad$ $\hat{v}^m =$

$\quad\quad \ln[\frac{1}{N}\sum_{n=1}^{N}(W_{t+\Delta t}^{m,n}(\pi_t^m))^{1-\gamma}exp(NN(t + \Delta t, \hat{X}_{t+\Delta t}^{m,n}, \hat{S}_{t+\Delta t}^{m,n}))] - (1 - \gamma)\ln[W_0]$
$\quad\quad$ *for* $m = 1...n_r$ ;

**13** $\quad$ Train a new network with input $(X_{T-\Delta t}^m, S_{T-\Delta t}^m)$ and output $\hat{v}^m$ and denote it by
$\quad\quad NN(t, X, S)$ ;

**14 while** $t = 0$ **do**

**15** $\quad$ Compute $\pi_0^*$ with with $NN(\Delta t, X, S)$ and Equation (10);

**16** $\quad$ Generate new paths of $S_t^z, X_t^z \quad for \quad z = 1...N_0$, use the estimation of
$\quad\quad$ transformed value function $NN(t, X, S)$ to compute $\pi_t^z$ and $W_T^z$.

**17** $\quad$ The expected utility is, $\hat{V}(0, W_0, S_0, X_0) = \frac{1}{N_0}\sum_{n=1}^{N_0} U(W_T^z)$

**18** return $\pi_0^*, \hat{V}(0, W_0, S_0, X_0)$

---

## 4. Application to 4/2 Model

Grasselli (2017) unified the 1/2 and 3/2 SV models and proposed the 4/2 SV model. The 4/2 model better captures the evolution of the implied volatility surface and uniformly bounds the instantaneous variance away from zero when weights on 1/2 and 3/2 factors are positive. We implement the NNMC on the 4/2 model and report the optimal allocation, expected utility and the annualized CER defined by

$$U(W_0(1 + CER)^T) = V(0, W_0, S_0, X_0) \tag{15}$$

Three versions of the 4/2 model are considered; all are specific cases of the generalized model (1). The first assumes market price of risk proportional to the volatility driver. In other words, the value function and the optimal allocation are solvable in closed form. The second incorporates stochastic jumps into the 4/2 model, while the last uses the preferred setting for the market price of risk in the economics/finance literature (i.e., proportional to the instantaneous volatility). The parameters used in this section are presented in Table 2 [1] and are estimated from the S&P 500 and its volatility index (VIX) in Cheng and Escobar-Anel (2021).

**Table 2.** Parameter values for 4/2 model.

| Parameter | Value | Parameter | Value |
|-----------|-------|-----------|-------|
| $T$ | 1 | $X_0$ | 0.04 |
| $r$ | 0.05 | $\lambda_S$ | 2.9428 |
| $\Delta_t^{re}$ | $\frac{1}{10}$ | $\Delta_t^{si}$ | $\frac{1}{60}$ |
| $S_0$ | 1.0 | $M_0$ | 1.0 |
| $W_0$ | 1 | $n_r$ | 100 |
| $N$ | 2000 | $N_0$ | 200000 |
| $\kappa_X$ | 7.3479 | $\theta_X$ | 0.0328 |
| $\sigma_X$ | 0.6612 | $a_s$ | 0.9051 |
| $b_S$ | 0.0023 | $\rho$ | $-0.7689$ |

### 4.1. A Solvable Case

Cheng and Escobar-Anel (2021) found the closed-form solution for an optimal dynamic portfolio when the stock price follows a 4/2 model with a market price of risk linear to the square root of the volatility driver $\sqrt{X_t}$. The dynamics of stock price $S_t$ and volatility driver $X_t$ are exhibited in (16):

$$\begin{cases} \frac{dM_t}{M_t} = rdt \\ \frac{dS_t}{S_t} = (r + \lambda_S(a_S X_t + b_S))dt + (a_S\sqrt{X_t} + \frac{b_S}{\sqrt{X_t}})dB_t^S \qquad <B_t^S, B_t^X> = \rho t \\ dX_t = \kappa_X(\theta_X - X_t)dt + \sigma_X\sqrt{X_t}dB_t^X. \end{cases} \tag{16}$$

Solving the associated Hamilton–Jacobi–Bellman (HJB) equation:

$$\begin{aligned} 0 = \sup_\pi \Big\{ & V_t + W_t(r + \lambda_S(a_S X_t + b_S)) + \kappa_X(\theta_X - X_t)V_X \\ & + \frac{1}{2}W_t^2\pi^2(a_S\sqrt{X_t} + \frac{b_S}{\sqrt{X_t}})^2 V_{WW} + \frac{1}{2}\sigma_X^2 X_t V_{XX} + \pi W_t(a_S X_t + b_S)\sigma_X\rho V_{WX} \Big\}, \end{aligned} \tag{17}$$

the optimal trading strategy and value function are given by

$$\begin{aligned} V(t, W, X) &= \frac{W^{1-\gamma}}{1-\gamma}e^{a(T-t)+b(T-t)X} \\ \pi_t^* &= \frac{X}{aX+b}\Big[\frac{\sigma_X\rho_{SX}b(T-t)}{\gamma} + \frac{\lambda_S}{\gamma}\Big]. \end{aligned} \tag{18}$$

The functions $a(T-t)$ and $b(T-t)$ are:

$$\begin{aligned} a(T-t) &= \gamma r(T-t) + \frac{2\kappa_X\theta_X}{k_2}\ln\frac{2k_3 e^{0.5(k_1+k_2)(T-t)}}{2k_3 + (k_1+k_3)(e^{k_3(T-t)}-1)} \\ b(T-t) &= \frac{k_0(e^{k_3(T-t)}-1)}{2k_3 + (k_1+k_3)(e^{k_3(T-t)}-1)}, \end{aligned} \tag{19}$$

with auxiliary parameters $k_0 = \frac{1-\gamma}{\gamma}\lambda_S^2$, $k_1 = \kappa_X - \frac{1-\gamma}{\gamma}\rho_{SX}\sigma_X\lambda_S$, $k_2 = \sigma_X^2 + \frac{(1-\gamma)\sigma_X^2\rho_{SX}^2}{\gamma}$ and $k_3 = \sqrt{k_1^2 - k_0 k_2}$.

The closed-form solution (see (18)) reveals that the value function in this case is an exponential linear function. Hence, we set the degree of polynomial to 1 when implementing NNMC with both the SEN and the IEN. Table 3 compares the optimal allocation, expected utility and CER from NNMC, the embedded PAMC and the theoretical solution. PAMC takes the least computational time. The optimal allocation obtained from PAMC is more accurate than the results from NNMC, while the differences in expected utility and CER are not significant. Furthermore, SEN slightly outperforms IEN in terms of the accuracy of

optimal allocation and computation efficiency. Moreover, the ReLU activation function is superior to the sigmoid and ELU function when the IEN is applied.

**Table 3.** Results for the 4/2 model with a market price of risk $\lambda_S \sqrt{X_t}$. We reported the optimal weights, expected utility and CER obtained with the theoretical result and with the approximation method for different levels of risk aversion $\gamma$. The standard deviation of estimated expected utility and CER from 100 runs is displayed in parentheses.

| | $\gamma = 2.0$ | $\gamma = 4.0$ | $\gamma = 6.0$ | $\gamma = 8.0$ | $\gamma = 10.0$ |
|---|---|---|---|---|---|
| **Theoretical** | | | | | |
| Weights ($\pi_0^*$) | 1.614 | 0.832 | 0.561 | 0.423 | 0.340 |
| Expected utility ($V_0^*$) | $-0.878$ | $-0.253$ | $-0.135$ | $-0.087$ | $-0.061$ |
| CER (%) | 13.85 | 9.62 | 8.15 | 7.40 | 6.95 |
| **PAMC** | | | | | |
| Weights ($\pi_0^{PAMC}$) | 1.615 | 0.833 | 0.561 | 0.423 | 0.340 |
| Relative error(%) | 0.001 | 0.05 | 0.05 | 0.04 | 0.04 |
| Expected utility ($V_0^{PAMC}$) | $-0.879$ (0.0005) | $-0.253$ (0.0002) | $-0.135$ (0.0001) | $-0.087$ (0.0001) | $-0.061$ (0.0001) |
| Relative error (%) | 0.04 | 0.04 | 0.05 | 0.06 | 0.08 |
| CER (%) | 13.80 (0.065) | 9.60 (0.033) | 8.14 (0.023) | 7.40 (0.017) | 6.95 (0.014) |
| Computational time (seconds) | 31.3 | 30.6 | 30.3 | 30.0 | 30.4 |
| **NNMC-SEN** | | | | | |
| Weights ($\pi_0^{SEN}$) | 1.612 | 0.831 | 0.560 | 0.422 | 0.339 |
| Relative error(%) | 0.15 | 0.18 | 0.20 | 0.22 | 0.23 |
| Expected utility ($V_0^{SEN}$) | $-0.879$ (0.0005) | $-0.253$ (0.0002) | $-0.135$ (0.0001) | $-0.087$ (0.0001) | $-0.061$ (0.0001) |
| Relative error (%) | 0.05 | 0.04 | 0.05 | 0.06 | 0.06 |
| CER (%) | 13.80 (0.065) | 9.60 (0.033) | 8.14 (0.026) | 7.40 (0.017) | 6.95 (0.014) |
| Computational time (seconds) | 56.4 | 57.2 | 57.6 | 57.0 | 57.9 |
| **NNMC-IEN (ReLU)** | | | | | |
| Weights ($\pi_0^{IEN\ ReLU}$) | 1.612 | 0.831 | 0.560 | 0.422 | 0.339 |
| Relative error(%) | 0.14 | 0.19 | 0.22 | 0.25 | 0.27 |
| Expected utility ($V_0^{IEN\ ReLU}$) | $-0.879$ (0.0005) | $-0.253$ (0.0002) | $-0.135$ (0.0001) | $-0.087$ (0.0001) | $-0.061$ (0.0001) |
| Relative error(%) | 0.05 | 0.04 | 0.05 | 0.06 | 0.06 |
| CER (%) | 13.80 (0.065) | 9.60 (0.033) | 8.14 (0.023) | 7.40 (0.017) | 6.95 (0.014) |
| Computational time (seconds) | 62.1 | 63.4 | 64.8 | 63.5 | 63.5 |
| **NNMC-IEN (sigmoid)** | | | | | |
| Weights ($\pi_0^{IEN\ sigmoid}$) | 1.612 | 0.831 | 0.560 | 0.422 | 0.339 |
| Relative error(%) | 0.15 | 0.20 | 0.23 | 0.27 | 0.27 |
| Expected utility ($V_0^{IEN\ sigmoid}$) | $-0.879$ (0.0005) | $-0.253$ (0.0002) | $-0.135$ (0.0001) | $-0.087$ (0.0001) | $-0.061$ (0.0001) |
| Relative error(%) | 0.05 | 0.04 | 0.05 | 0.10 | 0.10 |
| CER (%) | 13.80 (0.065) | 9.60 (0.033) | 8.14 (0.023) | 7.39 (0.017) | 6.94 (0.014) |
| Computational time (seconds) | 63.6 | 62.2 | 63.1 | 79.2 | 75.1 |
| **NNMC-IEN (ELU)** | | | | | |
| Weights ($\pi_0^{IEN\ ELU}$) | 1.612 | 0.831 | 0.560 | 0.422 | 0.339 |
| Relative error(%) | 0.16 | 0.19 | 0.19 | 0.28 | 0.31 |
| Expected utility ($V_0^{IEN\ ELU}$) | $-0.879$ (0.0005) | $-0.253$ (0.0002) | $-0.135$ (0.0001) | $-0.087$ (0.0001) | $-0.061$ (0.0001) |
| Relative error(%) | 0.06 | 0.10 | 0.05 | 0.18 | 0.20 |
| CER (%) | 13.78 (0.065) | 9.58 (0.034) | 8.14 (0.023) | 7.38 (0.017) | 6.93 (0.014) |
| Computational time (seconds) | 78.2 | 74.6 | 62.1 | 77.4 | 75.3 |

We repeat the estimation of expected utility (i.e., steps 14–16 in NNMC-SEN and steps 15–17 in NNMC-IEN) after the value function and optimal strategy are obtained. All approximation methods have similar standard deviations of the estimated expected utility

and CERs. Moreover, standard deviation decreases with an risk aversion level $\gamma$, which indicates that our approximation is more accurate for higher risk averse investors.

Figure 3 displays the expected utility and CER as a function of time to maturity $T$ when $\gamma = 2$. The expected utility increases with maturity $T$ as expected, while the CER decreases. Expected utility from PAMC, NNMC and the theoretical solution are visually the same. The comparison in portfolio performance is clearer by showing the CER: PAMC and NNMC produce CERs that are slightly smaller than the theoretical result. Furthermore, ELU seems to be inferior to the ReLU and sigmoid function, and the CER obtained from NNMC with the ELU activation function is slightly smaller than the results from other methods when the investment horizon is small.

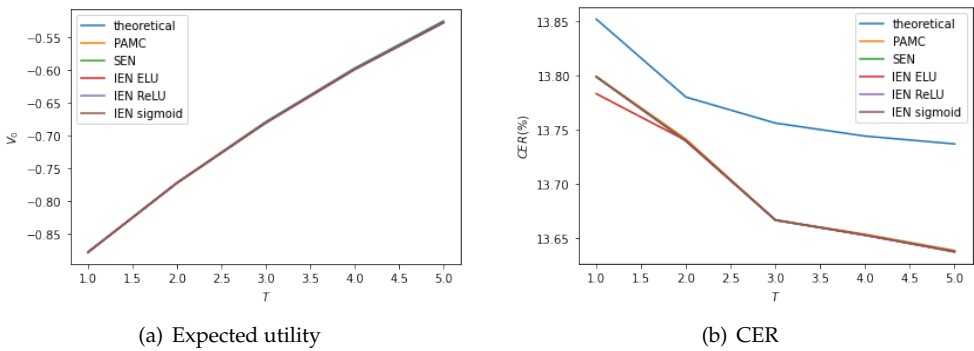

(a) Expected utility          (b) CER

**Figure 3.** $S_t$ follows the 4/2 model with a market price of risk $\lambda_S \sqrt{X_t}$, where (**a**) shows the expected utilities obtained with theoretical results and approximation methods versus investment horizon $T$; and (**b**) shows the CERs versus investment horizon $T$ given $\gamma = 2$.

### 4.2. An Unsolvable Case, 4/2 Model with Jumps

We then extended the 4/2 model to account for stochastic jumps. The dynamics of stock prices and volatility drivers are summarized by the SDE:

$$
\begin{cases}
\frac{dM_t}{M_t} = rdt \\
\frac{dS_t}{S_t} = (r + \lambda_S(a_S X_t + b_S) - \lambda_Q X_t \mu_N)dt + (a_S \sqrt{X_t} + \frac{b_S}{\sqrt{X_t}})dB_t^S + \mu_N dN_t & < B_t^S, B_t^X > = \rho t \\
dX_t = \kappa_X(\theta_X - X_t)dt + \sigma_X \sqrt{X_t}dB_t^X.
\end{cases}
\tag{20}
$$

Volatility and market price of risk are the same with the 4/2 model given in (16). $N_t$ is an independent Poisson process with intensity $\lambda_N X_t$, $\mu_N$ is the jump size, and $\lambda_Q X_t$ captures the market price of jump risk.

We used the set of jump risk parameters given in Liu and Pan (2003): $\lambda_N = \lambda_Q = 0.1/\theta_X$ and $\mu_N = 0.1$. Notably, the stock is expected to jump once every 10 years if $X_t$ stays at its mean level $\theta_X$. The degree of polynomial in PAMC and NNMC was chosen to be 1. In this case, the optimal strategy cannot be explicitly solved given the approximation of the value function at the next rebalancing time (see Propositions 1 and 2), which is therefore obtained by the Newton–Raphson method in NNMC. The optimal allocation, expected utility, CER obtained with NNMC and PAMC are reported in Table 4. When the stock follows the 4/2 model with jumps, PAMC is faster, followed by NNMC-SEN. Moreover, the accuracy of the estimated expected utility and CER from PAMC and NNMC are similar; the standard deviations of these approximation methods have little difference.

Figure 4 exhibits the expected utility and CER as a function of investment horizon $T$. Portfolios with a longer investment horizon are expected to achieve a better performance (i.e., higher expected utility) while CER decreases with $T$.

**Table 4.** Results for the 4/2 model with stochastic jumps. We report the optimal weights, expected utility and CER obtained via the approximation methods for different levels of risk aversion $\gamma$. The standard deviation of estimated expected utility and CER from 100 runs is displayed in parentheses.

| | $\gamma = 2.0$ | $\gamma = 4.0$ | $\gamma = 6.0$ | $\gamma = 8.0$ | $\gamma = 10.0$ |
|---|---|---|---|---|---|
| **PAMC** | | | | | |
| Weights ($\pi_0^{PAMC}$) | 1.545 | 0.797 | 0.537 | 0.405 | 0.325 |
| Expected utility ($V_0^{PAMC}$) | $-0.882$ (0.0006) | $-0.255$ (0.0003) | $-0.136$ (0.0002) | $-0.087$ (0.0001) | $-0.061$ (0.0001) |
| CER (%) | 13.43 (0.075) | 9.41 (0.039) | 8.01 (0.027) | 7.30 (0.020) | 6.87 (0.016) |
| Computational time (seconds) | 47.1 | 48.4 | 47.1 | 47.2 | 47.1 |
| **NNMC-SEN** | | | | | |
| Weights ($\pi_0^{SEN}$) | 1.545 | 0.797 | 0.537 | 0.405 | 0.325 |
| Expected utility ($V_0^{SEN}$) | $-0.882$ (0.0006) | $-0.255$ (0.0003) | $-0.136$ (0.0002) | $-0.087$ (0.0001) | $-0.061$ (0.0001) |
| CER (%) | 13.43 (0.075) | 9.41 (0.040) | 8.01 (0.027) | 7.30 (0.020) | 6.87 (0.016) |
| Computational time (seconds) | 74.2 | 77.3 | 72.5 | 72.3 | 82.7 |
| **NNMC-IEN (ReLU)** | | | | | |
| Weights ($\pi_0^{IEN\ ReLU}$) | 1.544 | 0.796 | 0.536 | 0.405 | 0.325 |
| Expected utility ($V_0^{IEN\ ReLU}$) | $-0.882$ (0.0006) | $-0.255$ (0.0003) | $-0.136$ (0.0002) | $-0.087$ (0.0001) | $-0.061$ (0.0001) |
| CER (%) | 13.43 (0.075) | 9.41 (0.040) | 8.01 (0.027) | 7.30 (0.020) | 6.87 (0.016) |
| Computational time (seconds) | 93.1 | 89.4 | 86.8 | 83.7 | 83.2 |
| **NNMC-IEN (sigmoid)** | | | | | |
| Weights ($\pi_0^{IEN\ sigmoid}$) | 1.544 | 0.796 | 0.537 | 0.404 | 0.324 |
| Expected utility ($V_0^{IEN\ sigmoid}$) | $-0.882$ (0.0006) | $-0.255$ (0.0003) | $-0.136$ (0.0002) | $-0.087$ (0.0001) | $-0.061$ (0.0001) |
| CER (%) | 13.43 (0.075) | 9.41 (0.040) | 8.01 (0.027) | 7.30 (0.020) | 6.87 (0.016) |
| Computational time (seconds) | 93.1 | 92.7 | 90.3 | 83.5 | 82.1 |
| **NNMC-IEN (ELU)** | | | | | |
| Weights ($\pi_0^{IEN\ ELU}$) | 1.544 | 0.796 | 0.537 | 0.404 | 0.325 |
| Expected utility ($V_0^{IEN\ ELU}$) | $-0.882$ (0.0006) | $-0.255$ (0.0003) | $-0.136$ (0.0002) | $-0.087$ (0.0001) | $-0.061$ (0.0001) |
| CER (%) | 13.43 (0.075) | 9.41 (0.040) | 8.01 (0.027) | 7.30 (0.020) | 6.87 (0.016) |
| Computational time (seconds) | 81.3 | 83.9 | 88.1 | 81.5 | 85.6 |

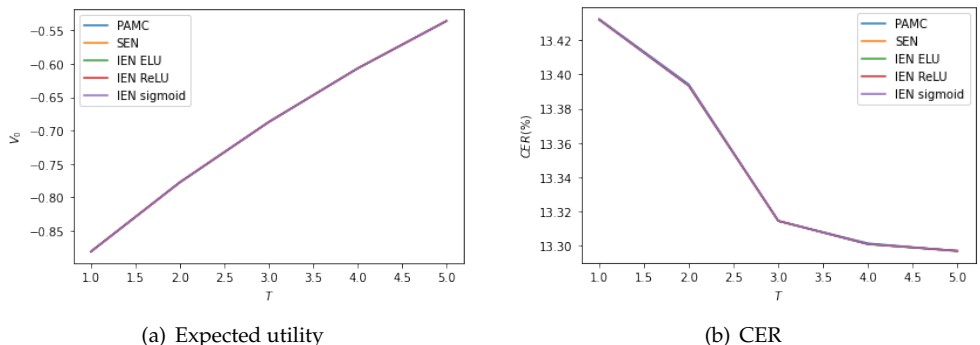

(a) Expected utility (b) CER

**Figure 4.** $S_t$ follows a 4/2 model with stochastic jump, where (**a**) shows the expected utilities obtained with the approximation methods versus investment horizon $T$; and (**b**) shows CERs versus investment horizon $T$ given $\gamma = 2$.

### 4.3. An Unsolvable Case, Market Price of Risk Proportional to Volatility

In this section, we consider an excess return, proportional to the instantaneous variance. The dynamics are given in (21), and a closed-form solution has not yet been found. We report the optimal allocation and expected utility from PAMC and NNMC, as well as

investigated the impact of maturity $T$. The degree of polynomial in PAMC and NNMC remains 1:

$$\begin{cases} \frac{dM_t}{M_t} = rdt \\ \frac{dS_t}{S_t} = (r + \lambda_S (a_S \sqrt{X_t} + \frac{b_S}{\sqrt{X_t}})^2)dt + (a_S \sqrt{X_t} + \frac{b_S}{\sqrt{X_t}})dB_t^S \quad < B_t^S, B_t^X >= \rho t \\ dX_t = \kappa_X (\theta_X - X_t)dt + \sigma_X \sqrt{X_t} dB_t^X. \end{cases} \quad (21)$$

Table 5 reports the optimal allocation, expected utility and CER from PAMC and NNMC. PAMC is still the most efficient method, followed by the NNMC-SEN. All methods achieve similar portfolio performance in terms of the expected utility and CER as well as the corresponding standard deviation. Figure 5 plots the expected utility and CER versus maturity $T$ when $\gamma = 2$, which further verifies the non-significant difference in expected utility and CER obtained from the methods.

**Table 5.** Results for the 4/2 model with a market price of risk $\lambda_S(a_S \sqrt{X_t} + \frac{b_S}{\sqrt{X_t}})$. We report the estimation of optimal weights, expected utility and CER obtained via approximations given different levels of risk aversion $\gamma$. The standard deviation of estimated expected utility and CER from 100 runs is displayed in parentheses.

| | $\gamma = 2.0$ | $\gamma = 4.0$ | $\gamma = 6.0$ | $\gamma = 8.0$ | $\gamma = 10.0$ |
|---|---|---|---|---|---|
| **PAMC** | | | | | |
| Weights ($\pi_0^{PAMC}$) | 1.539 | 0.789 | 0.531 | 0.400 | 0.321 |
| Expected utility ($V_0^{PAMC}$) | −0.882 (0.0005) | −0.255 (0.0002) | −0.136 (0.0001) | −0.087 (0.0001) | −0.061 (0.0001) |
| CER (%) | 13.38 (0.065) | 9.36 (0.033) | 7.97 (0.022) | 7.26 (0.017) | 6.84 (0.013) |
| Computational time (seconds) | 33.9 | 33.6 | 34.0 | 35.4 | 33.2 |
| **NNMC-SEN** | | | | | |
| Weights ($\pi_0^{SEN}$) | 1.537 | 0.788 | 0.530 | 0.399 | 0.320 |
| Expected utility ($V_0^{SEN}$) | −0.882 (0.0005) | −0.255 (0.0002) | −0.136 (0.0001) | −0.087 (0.0001) | −0.061 (0.0001) |
| CER (%) | 13.38 (0.065) | 9.36 (0.033) | 7.97 (0.022) | 7.26 (0.017) | 6.84 (0.013) |
| Computational time (seconds) | 62.7 | 62.6 | 62.4 | 62.7 | 62.9 |
| **NNMC-IEN (ReLU)** | | | | | |
| Weights ($\pi_0^{IEN\ ReLU}$) | 1.537 | 0.788 | 0.530 | 0.399 | 0.320 |
| Expected utility ($V_0^{IEN\ ReLU}$) | −0.882 (0.0005) | −0.255 (0.0002) | −0.136 (0.0001) | −0.087 (0.0001) | −0.061 (0.0001) |
| CER (%) | 13.38 (0.065) | 9.35 (0.033) | 7.97 (0.022) | 7.26 (0.017) | 6.84 (0.013) |
| Computational time (seconds) | 70.8 | 69.8 | 69.0 | 69.4 | 69.6 |
| **NNMC-IEN (sigmoid)** | | | | | |
| Weights ($\pi_0^{IEN\ sigmoid}$) | 1.537 | 0.788 | 0.530 | 0.399 | 0.320 |
| Expected utility ($V_0^{IEN\ sigmoid}$) | −0.883 (0.0005) | −0.255 (0.0002) | −0.136 (0.0001) | −0.088 (0.0001) | −0.061 (0.0001) |
| CER (%) | 13.38 (0.065) | 9.35 (0.033) | 7.97 (0.022) | 7.26 (0.017) | 6.84 (0.013) |
| Computational time (seconds) | 69.0 | 68.0 | 68.9 | 68.4 | 68.7 |
| **NNMC-IEN (ELU)** | | | | | |
| Weights ($\pi_0^{IEN\ ELU}$) | 1.537 | 0.788 | 0.530 | 0.399 | 0.320 |
| Expected utility ($V_0^{IEN\ ELU}$) | −0.882 (0.0005) | −0.255 (0.0002) | −0.136 (0.0001) | −0.087 (0.0001) | −0.061 (0.0001) |
| CER (%) | 13.38 (0.065) | 9.35 (0.033) | 7.97 (0.022) | 7.26 (0.017) | 6.84 (0.013) |
| Computational time (seconds) | 69.3 | 69.4 | 68.3 | 71.6 | 68.5 |

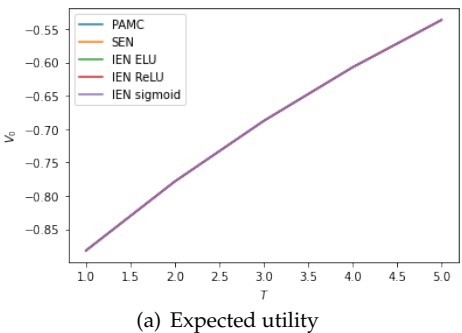

(a) Expected utility

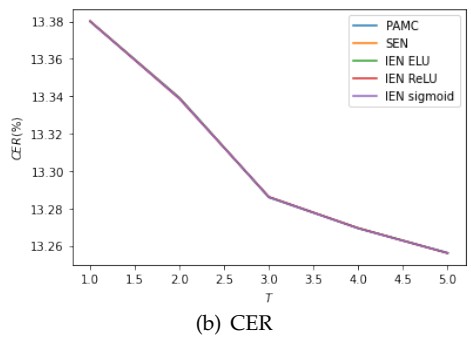

(b) CER

**Figure 5.** $S_t$ follows the 4/2 model with a market price of risk $\lambda_S(a\sqrt{X_t} + \frac{b}{\sqrt{X_t}})$, where (**a**) shows the expected utilities obtained with theoretical results and approximation methods versus investment horizon $T$; and (**b**) shows the CERs versus investment horizon $T$ given $\gamma = 2$.

## 5. Application to the OU 4/2 Model

Motivated by the 4/2 stochastic volatility model and mean-reverting price pattern popular among various asset classes (e.g., commodities, exchange rates, volatility indexes), Escobar-Anel and Gong (2020) defined an Ornstein–Uhlenbeck 4/2 (OU 4/2) stochastic volatility model for volatility index option and commodity option valuation. Equation (22) presents the dynamics involved in the OU 4/2 model, which is a specific case of (1) given $\theta(X_t, S_t) = (L_S + (\lambda_S - \frac{1}{2})(a_S\sqrt{X_t} + \frac{b_S}{\sqrt{X_t}})^2 - \beta_S \ln S_t)$, $\sigma(X_t, S_t) = (a_S\sqrt{X_t} + \frac{b_S}{\sqrt{X_t}})$, $a(X_t) = \kappa_X(\theta_X - X_t)$ and $b(X_t) = \sigma_X\sqrt{X_t}$. The parameters used in this section are reported in Table 6, which is estimated from the data of gold Exchange-traded fund (ETF) and the volatility index of gold ETF in Escobar-Anel and Gong (2020). There are two state variables in the OU 4/2 model; hence, the input in both the SEN and the IEN are 2. Furthermore, the degree of polynomial in PAMC and NNMC is 2:

$$\begin{cases} \frac{dM_t}{M_t} = rdt \\ \frac{dS_t}{S_t} = (L_S + \lambda_S(a_S\sqrt{X_t} + \frac{b_S}{\sqrt{X_t}})^2 - \beta_S \ln S_t)dt + (a_S\sqrt{X_t} + \frac{b_S}{\sqrt{X_t}})dB_t, \\ dX_t = \kappa_X(\theta_X - X_t)dt + \sigma_X\sqrt{X_t}dB_t^X \\ <dB_t, dB_t^X> = \rho dt. \end{cases} \tag{22}$$

**Table 6.** Parameter value for the OU 4/2 model.

| Parameter | Value | Parameter | Value |
|:---:|:---:|:---:|:---:|
| $T$ | 1 | $X_0$ | 0.04 |
| $r$ | 0.05 | $\lambda_S$ | 0.572 |
| $\Delta_t^{re}$ | $\frac{1}{60}$ | $\Delta_t^{si}$ | $\frac{1}{60}$ |
| $S_0$ | 120.0 | $M_0$ | 1.0 |
| $W_0$ | 1 | $n_r$ | 100 |
| $\kappa_X$ | 4.7937 | $\theta_X$ | 0.0395 |
| $\sigma_X$ | 0.2873 | $a_S$ | 1 |
| $b_S$ | 0.002 | $\rho$ | $-0.08$ |
| $L$ | 3.7672 | $\beta_S$ | 0.78 |
| $N$ | 2000 | $N_0$ | 200,000 |

SEN performs worse than IEN when fitting the value function with the OU 4/2 model. Sometimes, SEN significantly deviates from the true value function, which results in poor portfolio performances and the occurrence of negative terminal wealth. Therefore, we excluded the results from NNMC-SEN in this section. Table 7 compares the optimal allocation, expected utility and CER obtained for the OU 4/2 model. PAMC and NNMC-IEN produce similar optimal allocations, both outperforming NNMC-SEN. Furthermore, we also estimated the standard deviation of expected utility and CER, which demonstrates that NNMC leads to a less volatile estimation of expected utility and CER than PAMC in most cases. In contrast to the results for the 4/2 model, IEN is more efficient than SEN. We conclude that IEN is suitable for the model with a complex structure and multiple state variables. The expected utility and CER as a function of the maturity $T$ when $\gamma = 2$ is plotted in Figure 6. Both the expected utility and CER increase with $T$. The expected utility and CER obtained from PAMC and NNMC-IEN visually overlap and are slightly higher than that of NNMC-SEN. Moreover, the selection of activation function in IEN makes little difference.

**Table 7.** Results for the OU 4/2 model. We report the estimation of optimal weights, expected utility and CER obtained via approximations for different levels of risk aversion $\gamma$. The standard deviation of estimated expected utility and CER from 100 runs is provided in parentheses.

| | $\gamma = 2.0$ | $\gamma = 4.0$ | $\gamma = 6.0$ | $\gamma = 8.0$ | $\gamma = 10.0$ |
|---|---|---|---|---|---|
| **PAMC** | | | | | |
| Weights ($\pi_0^{PAMC}$) | 0.068 | 0.026 | 0.015 | 0.010 | 0.008 |
| Expected utility ($V_0^{PAMC}$) | −0.888 (0.0006) | −0.255 (0.0003) | −0.136 (0.0002) | −0.087 (0.0002) | −0.061 (0.0001) |
| CER (%) | 12.65 (0.073) | 9.28 (0.047) | 8.00 (0.035) | 7.32 (0.028) | 6.90 (0.024) |
| Computational time (seconds) | 103.9 | 104.6 | 104.4 | 104.5 | 104.3 |
| **NNMC-SEN** | | | | | |
| Weights ($\pi_0^{SEN}$) | 0.134 | 0.056 | 0.042 | 0.040 | 0.029 |
| Expected utility ($V_0^{SEN}$) | −0.888 (0.0006) | −0.256 (0.0003) | −0.136 (0.0002) | -0.087 (0.0001) | −0.061 (0.0001) |
| CER (%) | 12.62 (0.076) | 9.26 (0.045) | 7.97 (0.032) | 7.29 (0.025) | 6.87 (0.020) |
| Computational time (seconds) | 439.5 | 477.5 | 434.3 | 446.9 | 449.7 |
| **NNMC-IEN (ReLU)** | | | | | |
| Weights ($\pi_0^{IEN\ ReLU}$) | 0.070 | 0.028 | 0.016 | 0.011 | 0.007 |
| Expected utility ($V_0^{IEN\ ReLU}$) | −0.888 (0.0006) | −0.255 (0.0003) | −0.136 (0.0002) | −0.087 (0.0002) | −0.061 (0.0001) |
| CER (%) | 12.65 (0.072) | 9.29 (0.045) | 8.00 (0.033) | 7.32 (0.026) | 6.90 (0.022) |
| Computational time (seconds) | 190.3 | 190.6 | 190.4 | 187.8 | 185.1 |
| **NNMC-IEN (sigmoid)** | | | | | |
| Weights ($\pi_0^{IEN\ sigmoid}$) | 0.067 | 0.026 | 0.015 | 0.010 | 0.007 |
| Expected utility ($V_0^{IEN\ sigmoid}$) | −0.888 (0.0006) | −0.255 (0.0003) | −0.136 (0.0002) | −0.087 (0.0001) | −0.061 (0.0001) |
| CER (%) | 12.65 (0.072) | 9.28 (0.044) | 8.00 (0.033) | 7.32 (0.026) | 6.90 (0.022) |
| Computational time (seconds) | 185.7 | 186.0 | 185.2 | 181.4 | 181.9 |
| **NNMC-IEN (ELU)** | | | | | |
| Weights ($\pi_0^{IEN\ ELU}$) | 0.072 | 0.031 | 0.015 | 0.010 | 0.008 |
| Expected utility ($V_0^{IEN\ ELU}$) | −0.888 (0.0006) | −0.255 (0.0003) | −0.136 (0.0002) | −0.087 (0.0001) | −0.061 (0.0001) |
| CER (%) | 12.65 (0.072) | 9.28 (0.044) | 8.00 (0.033) | 7.32 (0.026) | 6.90 (0.022) |
| Computational time (seconds) | 185.6 | 184.1 | 188.1 | 195.1 | 193.7 |

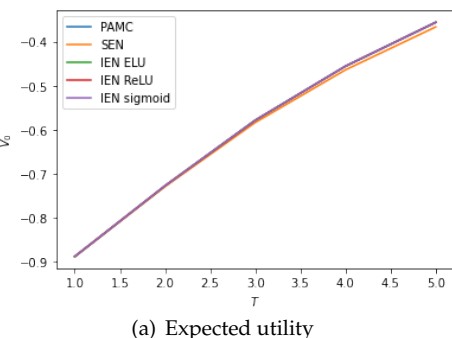
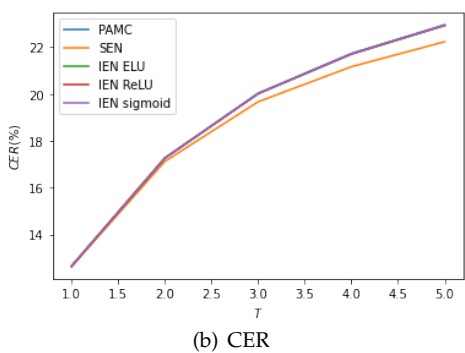

(a) Expected utility

(b) CER

**Figure 6.** $S_t$ follows the OU 4/2 model, where (**a**) shows the expected utilities obtained via approximation methods versus investment horizon $T$; and (**b**) shows the CERs versus investment horizon $T$ given $\gamma = 2$.

## 6. Conclusions

This paper investigated fitting the value function in an expected utility, dynamic portfolio choice using a deep learning model. We proposed two architectures for the neural network, which extends the broadest solvable family of value functions (i.e., the exponential polynomial function). We measured the accuracy and efficiency of various types of NNMC methods on the 4/2 model and the OU 4/2 model. The difference in optimal allocation, expected utility and CER is insignificant when the stock price follows the 4/2 model. The embedded PAMC is superior to NNMC due to the lower parametric space, hence its efficiency. Furthermore, when considering the OU 4/2 model, NNMC-SEN is inferior to a polynomial regression (PAMC) and to the NNMC-IEN in terms of expected utility and CER.

In summary, NNMC benefits from the popular exponential polynomial representation (embedded PAMC method) to propose a network architecture flexible enough to reach beyond affine models. Although the best setting, NNMC-IEN (ELU), is not as efficient as PAMC, neural networks demonstrate the way to tackle more advanced models along the lines of Markov switching, Lévy processes and fractional Brownian processes.

**Author Contributions:** The authors contributed equally. All authors have read and agreed to the published version of the manuscript.

**Funding:** This research was funded by NSERC, grant number RGPIN-2020-05068.

**Conflicts of Interest:** The authors declare no conflict of interest.

## Note

1    $\Delta_t^{re}$ is the portfolio rebalancing interval, $\frac{1}{\Delta_t^{re}}$ indicates the rebalancing frequency. The Euler method with step size $\Delta_t^{si}$ is applied in generating the stock price and states variables.

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
