# Peer review of "A Neural Network Monte Carlo Approximation for Expected Utility Theory"

_jrfm, doi:10.3390/jrfm14070322_

Round 1
Reviewer 1 Report
The paper studies a continuous portfolio optimization problem with stochastic volatility; Monte Carlo simulations are performed, with help of neural networks. I think the paper can be accepted after a few minor revisions.
- Particular focus is made on the 4/2 SV model, but I wonder if the methodology would be easily transferable to models with other elasticity constant, for instance a 3/2 model (see e.g. Carr and Sun, Rev. Deriv. Res. 2007).
- As Monte Carlo methods are involved, it would be interesting to have a bit more practical details such as convergence speed or confidence intervals.
- Also, as a non specialist of deep learning techniques, I wonder whether neural networks provide better variance reduction than classical techniques (such as antithetic variates), or simply a faster convergence? Maybe a short discussion on the topic would be welcome.
- P.15 please correct "Levy" to "Lévy".
Author Response
Please see the file attached.
Thank you.
Reviewer 2 Report
Please, see the file attached

Author Response

(The authors gave the same response as above.)

Round 2
Reviewer 2 Report
After reading the new version of the manuscript, I notedd a great improvement of the work. The authors have enriched both the theoretical part and the numerical exercise according to all the suggestions I proposed.
Hence, I believe that the paper is now ready for publication.